# Safe Procedure for Efficient Hydrodynamic Gene Transfer to Isolated Porcine Liver in Transplantation

**DOI:** 10.3390/ijms25031491

**Published:** 2024-01-25

**Authors:** Luis Sendra, Mireia Navasquillo, Eva M. Montalvá, David Calatayud, Judith Pérez-Rojas, Javier Maupoey, Paula Carmona, Iratxe Zarragoikoetxea, Marta López-Cantero, María José Herrero, Salvador F. Aliño, Rafael López-Andújar

**Affiliations:** 1Pharmacogenetics and Gene Therapy Unit, La Fe Health Research Institute, 46026 Valencia, Spain; luis.sendra@uv.es (L.S.); maria.jose.herrero@uv.es (M.J.H.); 2Gene Therapy and Pharmacogenomics, Department of Pharmacology, University of Valencia, 46010 Valencia, Spain; 3Department of HPB Surgery and Transplantation Unit, Division of General Surgery, University and Polytechnic La Fe Hospital, 46026 Valencia, Spain; 4Hepatology, HBP Surgery and Transplants Group, La Fe Health Research Institute, 46026 Valencia, Spain; 5Network Biomedical Research Center for Liver and Digestive Diseases, CIBERehd, Health Institute Carlos III, 28029 Madrid, Spain; 6Pathology Department, University and Polytechnic La Fe Hospital, 46026 Valencia, Spain; 7Anesthesia and Resuscitation Service, University and Polytechnic La Fe Hospital, 46026 Valencia, Spain

**Keywords:** gene therapy, interleukin-10, cytokines, naked DNA

## Abstract

Although calcineurin inhibitors are very effective as immunosuppressants in organ transplantation, complete graft acceptance remains as a challenge. Transfer of genes with immunosuppressant functions could contribute to improving the clinical evolution of transplantation. In this sense, hydrodynamic injection has proven very efficacious for liver gene transfer. In the present work, the *hIL-10* gene was hydrofected ‘ex vivo’ to pig livers during the bench surgery stage, to circumvent the cardiovascular limitations of the procedure, in a model of porcine orthotopic transplantation with a 10-day follow-up. We used IL-10 because human and porcine proteins can be differentially quantified and for its immunomodulatory pleiotropic functions. Safety (biochemical parameters and histology), expression efficacy (RNA transcription and blood protein expression), and acute inflammatory response (cytokines panel) of the procedure were evaluated. The procedure proved safe as no change in biochemical parameters was observed in treated animals, and human IL-10 was efficaciously expressed, with stationary plasma protein levels over 20 pg/mL during the follow-up. Most studied cytokines showed increments (interferon-α, IFN-α; interleukin-1β, IL-1β; tumor necrosis factor α, TNFα; interleukin-6, IL-6; interleukin-8, IL-8; interleukin-4, IL-4; and transforming growth factor-β, TGF-β) in treated animals, without deleterious effects on tissue. Collectively, the results support the potential clinical interest in this gene therapy model that would require further longer-term dose–response studies to be confirmed.

## 1. Introduction

The hydrodynamic procedure for gene transfer consists of the rapid administration of a large volume of saline aqueous solution containing the gene of interest at defined concentrations. This strategy presents the advantage that DNA is administered without any carrier, and then neither toxicity, which is more common with viral vectors [1], nor interactions can occur, and the gene release in the target is not limited [2]. As disadvantages, the gene is delivered without targeting, so it could potentially access any cell, and the volume required for efficient gene transfer could compromise the patient’s hemodynamics. The strategy developed in previous works [3] and the present one for loco-regional administration of gene solution in vascular isolated organs permitted circumvention, partially at least, of these disadvantages since the DNA was injected only in the target organ and the conditions for administration could be moderated. Furthermore, the technique would permit the administration of the gene using minimally invasive catheterization, making its clinical translation feasible [4].

Hydrodynamic injection of naked DNA has been proven to be a safe and efficacious procedure for gene transfer in different organs, especially in the liver. In the early 1990s, it was reported for the first time that DNA could be transferred into the mouse liver by means of its direct injection [5,6]. In 1999, Liu et al. [7] and Zhang et al. [8] described hydrodynamic gene transfer consisting of the rapid injection of a large saline solution volume containing the DNA of interest through the tail vein in mice. Several groups and ours participated in the development of this technique to improve the efficacy of gene transfer and expression. The work of our group, Aliño et al. [9], reported for the first time sustained therapeutic levels of human alpha-1-antitrypsin protein (>1 mg/mL) for periods longer than 6 months after injecting a plasmid containing this entire gene in a murine model of hydrodynamic gene transfer in 2003. Since these promising results, great effort has been made in order to translate this technique to a clinical setting. In this process, different larger animal models have been employed and the conditions of gene injection have been adapted to permit exerting the pressure mediated by the hydrodynamic procedure without affecting the hemodynamic status of the entire body. In this sense, different researchers have proposed distinct pig models [10,11,12,13]. The reported results agreed with the suggestion of needing to pressurize the target organ by isolating it from the bloodstream. Our group participated in the translational process employing different porcine models, aiming to exclude the liver vasculature completely or partially, and evaluated the efficiency of gene transfer and decoding. In this regard, we described three models for hydrodynamic gene transfer in the pig liver: open non-invasive catheterization (only one balloon catheter for gene administration in a single lobe with closed backflow) [14]; closed catheterization (three balloon catheters for targeting the whole liver and excluding the entire venous circulation) [15]; and surgery-closed (complete laparotomy for identifying both arterial and venous vessels and targeting the whole watertight liver) [16,17]. We observed that the more the liver was pressurized, the higher the efficacy of expression of the protein encoded by the transferred gene. However, the liver pressurization forced the employment of more invasive procedures.

The porcine model permitted evaluating the potential of this strategy to efficiently transfer a naked entire gene to the liver with enough bioavailability to be decoded and express high levels of the protein of interest. Recently, Kamimura et al. [18] reported efficient gene transfer in the baboon liver by lobe-directed hydrodynamic injection employing IX coagulation factor. Aiming to determine whether this would occur in human beings, we developed a procedure to transfer a gene using the hydrodynamic procedure in watertight human liver segments derived from surgical resection in patients with cancer [19]. After gene transfer, tissue samples were cultured and the process of gene decoding was quantified at different time points. This study proved that, despite the fact that the liver had been removed from a patient and had no nutrients and oxygen supply, human liver tissue could efficaciously express the gene for some days.

The procedure demonstrated its potential to efficiently transfer genes to the porcine and human liver; however, the need for organ pressurization and the large volumes injected (≥200 mL) in a short time (approx. 20 s) made its clinical application unacceptable. For this reason, we decided to test it in a clinical setting that could benefit from this procedure without affecting the patient. In this respect, liver transplantation offers an opportunity for safe gene transfer in a watertight organ during the bench surgery step between organ collection and its implantation in the recipient. In this stage, surgeons have access to the entire vasculature [20]. Hence, the liver could be safely pressurized and the gene hydrodynamically transferred to the entire organ without affecting either the regular performance of organ transplantation or the hemodynamic status of the patient.

One of the main limitations of liver transplantation is immunosuppression. Despite the substantial efficacy of calcineurin inhibitors, post-transplantation early mortality remains as high as 5–12%, and acute cellular rejection appears within the first year post-intervention in 25–30% of patients [21] or more [22]. For this reason, it was decided to employ a gene that encoded a protein with immunomodulatory function, interleukin-10, instead of a tracer gene, GFP, since this cytokine plays a central role in the modulation of immune activation through different pleiotropic functions. Interleukin-10 (IL-10) is a pleiotropic cytokine that plays a fundamental role in modulating inflammation and maintaining cell homeostasis. It primarily acts as an anti-inflammatory cytokine, protecting the body from an uncontrolled immune response. Given the pivotal role of IL-10 in immune modulation, this cytokine could have relevant implications in pathologies characterized by the hyperinflammatory state. IL-10 leads to the shutdown of the inflammatory immune response, both directly, by the suppression of macrophage and dendritic cell activity, and indirectly, by limiting T cell activation, differentiation, and effector function, and promoting peripheral tolerance [23].

This molecule was reported to improve the outcome in transplantation [24] and polymorphisms in its gene have been described to influence transplantation success. Additionally, we selected it because it could also serve as tracer molecule at the RNA and protein levels to permit differential quantification of the expression of the exogenous human gene. It has been reported that the amino acid sequence homology between human and mouse, rabbit, pig, and cow IL-10 is high. Human and porcine IL-10 proteins present 76% homology, more than that observed between human and mouse proteins. Human IL-10 cDNA presents important homology with pig IL-10, with 82% homology in the nucleotide sequence [25]. Moreover, IL-10 has been shown to be functionally active across species barriers, including human, mouse, rabbit, pig, and cow [26,27,28].

In the present work, an orthotopic liver transplantation porcine model with standard immunosuppressant treatment was employed to hydrodynamically transfer the human IL-10 gene to the graft liver during bench surgery, as previously described [14,15,16,17,19]. Our main goal was to evaluate the aspects related to the pharmacokinetics of the procedure of gene delivery to mediate protein expression in the blood and the further potential effect of IL-10 in the immediate post-transplantation expression of cytokines for 10 days.

Collectively, the procedure proved to be safe and permitted the efficient expression and plasma release of human IL-10 protein, achieving sustained levels during the entire follow-up with potential immunosuppressive activity without any damage with respect to control group animals.

## 2. Results

### 2.1. Biochemical Variables and Liver Function

Although differences were observed between the control and treated groups at some time points, none of the determined parameters (Table 1; Figure 1) showed statistically significant differences between groups. In spite of these differences, the observed levels did not exceed safety values in any case and tended to normalize after 1 day post-surgery.

### 2.2. Human IL-10 Protein Concentration in Plasma

Human IL-10 protein (Figure 2) in treated animals achieved stable plasma concentrations over 20 pg/mL until day 10, with a near 40 pg/mL peak on day 2, which was higher (*p* = 0.012; *t*-test) than in control animals, in which concentrations remained around the detection threshold during the entire follow-up.

### 2.3. Human IL-10 RNA Expression in Liver Tissue Samples

Figure 3 shows the tissue amount of human IL-10 RNA in the different areas of the liver, expressed as copy number per cell after subtracting the background read. All liver lobes presented detectable expression of *hIL-10* mRNA, although the yield was different depending on the area of the organ. Whereas both medial and lateral segments of the right lobes showed similar intermediate transcription rates, in the left lobes, great differences between the medial and lateral areas were detected, the latter showing the lowest efficacy. The proximal area of the medial left lobe showed the highest production of mRNA, more than 1000-fold higher than in the lateral left segments. We think that the reason for such differences must be the anatomical structure of the suprahepatic vasculature that orientates the catheter in that direction and favors the irrigation of those areas.

### 2.4. Expression of Porcine Interleukin-10

The *hIL-10* liver gene transfection did not alter the expression of natural porcine IL-10 and, thus, no difference between the treated and control groups (*p* = 0.393) was observed (Figure 4).

### 2.5. Plasma Levels of Other Porcine Cytokines

Several cytokines involved in the immune response were quantified in blood samples. Except for interferon gamma and pIL-10, levels of all determined cytokines were higher in the DNA-transfected group. Levels of tumor necrosis factor-α (TNF-α), interleukin-1β (IL-1β), interleukin-4 (IL-4), interleukin-6 (IL-6), and interleukin-8 (IL-8) were higher from the beginning of sample collection (1h) in transfected animals and their levels were sustained during the entire follow-up (Figure 5A), with significantly different behavior (*p* = 0.0017, *p* = 0.0141, *p* = 0.0054, *p* = 0.0046, and *p* = 0.0057, respectively). Interferon-α (IFN-α) and interferon-γ (IFN-γ) presented similar expression profiles in both groups (Figure 5B). Although it was not significant (*p* = 0.089), the expression of IFN-α increased in the transfected group during the first 2 days, maybe due to the administration of double-stranded DNA, as previously reported. Nucleic acids derived from pathogens or released from damaged cells can activate the immune system. TLR9, a membrane-bound DNA sensor, detects CpG DNA at the endosome to induce type I IFN production. There are also TLR9-independent pathways that can recognize double-stranded DNA (dsDNA) in the cytoplasm, preferentially mediating the production of IFN-α but not IL-12 [29,30]. Interleukin-12 (IL-12) expression levels in transfected pigs were significantly higher (*p* = 0.0001) during the follow-up, and this could be due to an immune response against the presence of ssRNA in the bloodstream, which could happen since the administered plasmid had already been proven to express *hIL-10* mRNA in few hours [17,19]. RNA-sensing mechanisms are mediated by Toll-like receptors (TLR) and by retinoic acid-inducible gene I-like receptors (RLRs), which induce the secretion of IL-12 but not IFN-α [29,30]. Plasma levels of immunomodulatory cytokine TGF-β (transforming growth factor-β) were significantly higher (*p* = 0.0323) in transfected animals during the entire follow-up.

### 2.6. Histological Evaluation of Liver Injury

In spite of immunosuppressant treatment, we observed three cases of acute rejection, two in the treated group (one of them very incipient and light) and another one in the control group. To precisely define the liver acute rejection, we classified the different types and level of liver injury observed in the histologic preparations. In Figure 6A, representative images of the general injury signs observed, such as ischemic injury and unspecific inflammation, are shown. In Figure 6B, acute rejection signs based on Banff parameters, such as duct injury, portal inflammation, and venular inflammation, can be observed.

To differentiate the potential effects of IL-10 modulatory gene injection, a pathologist assigned quantitative severity values (0–3) to the injury effects observed in the samples of each animal. No statistical difference in general damage was observed between the control and treated groups. Although rejection appeared in one only animal per group, the severity of the overall score of acute rejection signs (Figure 7) was higher in the control pig (7.5 vs. 4.25, *p* = 0.043).

## 3. Discussion

The aim of this study was to evaluate the efficiency of the hydrodynamic injection procedure to mediate hepatic *hIL-10* gene transfer with bioavailability for decoding, express the protein in the bloodstream in a model of liver transplantation, and evaluate the acute effect on cytokine release in the immediate post-transplantation stage. Following gene transfection, transcription of *hIL-10* RNA was distributed throughout all liver lobes (0.1 to 1000 copies per cell, depending on the area) and sustained levels of human IL-10 protein over basal functional levels were observed (20 pg/mL) during the follow-up period, indicating successful gene transfer. Interestingly, we observed overall increments in both anti- and pro-inflammatory cytokines, which did not correlate with more severe liver injury.

The *hIL-10* gene was selected since it could serve as a tracer due to the ability to differentiate porcine and human proteins, interest in its use in liver transplantation, and our previous knowledge of its physiology and expression profile. The plasmid was injected under optimal conditions, previously established by our research group, into a watertight liver on the surgical bench immediately before implantation. No statistical differences were found in plasma levels of biochemical parameters, except for albumin and liver enzymes, suggesting that the gene transfer did not affect normal liver function.

In the treated animals, *hIL-10* concentrations reached a peak of 40 pg/mL on day 2, followed by a decrease to 20 pg/mL, which remained stable until day 10. This indicated that the plasmid successfully reached the cells, underwent transcription to RNA, translation to protein, and release into the bloodstream. These levels were expected to mediate an immunomodulatory effect, considering that the calculated half-maximal inhibitory concentration (IC50) of IL-10 for TNF-α is 124 pg/mL [31]. The local concentration achieved in the liver environment was likely higher, highlighting the potential clinical relevance of this procedure. Notably, the porcine IL-10 plasma levels (100 pg/mL) in both groups were higher than those of the human protein (20–40 pg/mL). We assume that the effect must have been induced by the increase in human protein concentration, since although we cannot rule out the possibility of a synergistic effect between human and porcine interleukin-10 on the expression of other cytokines, we only observed differences in human IL-10 expression between the control and treated groups.

Despite the increased and sustained expression of both pro- and anti-inflammatory cytokines after *hIL-10* transfection, the hIL-10-treated animals generally displayed better wellbeing, as observed through their behavior and response to stimuli. In accordance with the levels of analyzed cytokines, one would anticipate that pigs treated with IL-10 would exhibit a greater and sustained inflammatory response throughout the entire study duration. Mediators conventionally reported as pro-inflammatory and, consequently, of potential risk for transplant evolution [32] (IL-1, IL-6, IL-8, TNF-α, IL-12–IL-23) maintained higher levels throughout the analysis period in the treated pigs compared to the controls, where they normalized by days 2–4 post-intervention. However, assessment of anatomopathological parameters indicated that the severity of Banff parameters, used to define the onset of rejection, was significantly higher (*p* = 0.0426) in the case in the control group. Conversely, levels of proteins with potential anti-inflammatory roles (IL-10, TGF-β) were also elevated in the treated pigs, which could favor the establishment of tolerance. These findings would require confirmation in longer-term studies wherein the Treg response could be assessed. These pleiotropic cytokines are known to participate in various activation or modulation [33,34] pathways during the acute phase of inflammation. Aiming to understand this pleiotropic response in the acute phase, we summed up the available information in Figure 8, where the signaling of interleukins, antigen activation, and effects of the immunosuppressant drug, tacrolimus, are represented as responsible pathways for the final activating or suppressant response.

In our study, immune activation mediated by MHC antigen recognition initiates the expression of IL-2 and clonal cell expansion. This cytokine activates its own receptor, promoting T cell proliferation, also including the generation of immunosuppressive Treg cells through the expression of Foxp3 [35]. However, calcineurin inhibitors block the production of IL-2, limiting the establishment of late immune tolerance mediated by Tregs [36]. Interleukin-10 (IL-10) is a pleomorphic cytokine produced by most activated immune cells, including B cells, mast cells, granulocytes, macrophages, dendritic cells, and multiple T cell subsets such as Tregs. Receptor ligation activates JAK/STAT3 signaling, leading to changes in the expression of immunomodulatory genes, such as Foxp3, inhibiting pro-inflammatory mediators, decreasing antigen presentation, and enhancing the tolerance functions of these cells. IL-10 can act as a feedback regulator that affects the control and resolution of inflammation via autocrine and paracrine mechanisms. In addition, IL-10 is thought to inhibit apoptotic signaling pathways, such as the p38 MAPK (mitogen-activated protein kinase) pathway, and maintain and expand Treg cell populations [37]. On the other hand, IL-10 can positively enhance activation and proliferation of certain immune cell types, including mast cells, CD8+ T cells, NK cells, and B cells, although the molecular mechanisms and functional consequences of such activity remain to be elucidated [38].

In this regard, we observed that IL-10 gene transfection increased the expression of IL-6 (STAT3) and TGF-β (SMAD2/3) [39], which we conjecture may compensate for the decrease in Foxp3 expression through different pathways than those employed by IL-2 (STAT3/5) [40]. Additionally, interleukin-10 could directly mediate the maintenance of Tregs [41] and indirectly induce the expression of TGF-β [39,42], although this should be confirmed in further longer-term dose-response studies. The direct action of IL-10 on Tregs through the Jak1/STAT3 signaling pathway could be responsible for enhancing the autocrine production of IL-10, and this could control the feedback response and possibly the expansion of Treg cells [43].

## 4. Materials and Methods

### 4.1. Animals

All animals received human care according to the “Guide for the Care and Use of Laboratory Animals”. Female 3-month-old pigs (25–30 kg) were individually housed in pigsties. Anesthesia was induced with ketamine (5–10 mg/kg, IM), midazolam, (0.3 mg/kg, IM), and propofol (4–6 mg/kg, IV). For its maintenance, inhaled sevoflurane was used (2%). All pigs were intubated for mechanical ventilation. Intra-intervention analgesia was achieved with fentanyl (10 µg/kg, IV, each 30 min) and buprenorphine (0.02 mg/kg, IV). The latter was also employed for post-surgery control of pain (0.15 mg/12 h until day 4, then it was reduced by half or removed). If fever appeared, intravenous meloxicam was scheduled. In the recipients, a central 3-way catheter was placed in the jugular vein for drug administration and blood samples collection, and an intra-arterial one was inserted in the femoral artery for pressure monitoring and intraoperative blood sampling. Antibiotic (amoxicillin/clavulanic acid, 250 mg, IV) was administered during the intervention and each day during the stabling period. All pigs, control (*n*: 5) and treated (*n*: 5), received immunosuppressant treatment as usual. These drugs were tacrolimus (0.04 mg/kg, per day, IV; this could be switched to oral Advagraf^®^ (Astellas pharma Europe, Addlestone, UK) if parenteral intolerance appeared) and methylprednisolone in a 6-day tapering dosage regimen (500 mg intra-intervention, 125 mg, 100 mg, 75 mg, 50 mg, and 25 mg, per day, IV). Electrocardiogram, oxygen saturation, arterial and venous pressures, and PEtCO_2_ were also monitored during the procedure. Blood gas determinations were performed during dissection and the anhepatic and reperfusion stages to evaluate the condition of the animal and adjust the administration of drugs and fluids. From day 1 onwards, omeprazole was administered (20 mg/day, IV) to avoid potential drug-derived gastric damage. The experiments were approved by the Animal Biological Research Ethics Committee of Hospital La Fe (ref. 2018/VSC/PEA/0145; Date of approval 2 July 2018).

### 4.2. Surgical Procedure

#### 4.2.1. Donor

After anesthesia induction and hemodynamic control, the surgical procedure was carried out as described by Fondevila et al. [44] with few modifications and the follow-up period was doubled. In brief, a midline xiphopubian laparotomy was performed. Then, the liver pedicle elements (bile duct, portal vein, and hepatic artery) were located and dissected as distally as possible. Bile duct was flushed with 100 mL saline solution; intravenous sodium heparin (3 mg/kg) was administered. After 3 min, the portal vein and aorta artery were cannulated and the liver was perfused with cold Celsior^®^ (Institute Georges Lopez, IGG, Lissieu, France) preservation solution, and it was exsanguinated by an excision in the inferior vena cava. After cold perfusion, complete hepatectomy was performed.

#### 4.2.2. Bench Surgery and Gene Transfer

Immediately after the hepatectomy, the organ was prepared on the bench. The inferior vena cava was sectioned at the diaphragm, and all branches of the celiac arterial trunk and the portal vein were preserved for as long as possible. In all livers, cholecystotomy was performed and the cystic duct was resected. Then, the liver was cold-preserved in Celsior^®^ at 4 °C. In the case of control pigs, no further intervention was required until the graft implant. In *hIL-10* plasmid-treated pigs, the gene was hydrodynamically transferred as follows: the inferior vena cava was clamped in both extremes, the portal vein and liver artery were also closed, and the gene (20 µg/mL in 200 mL saline solution) was retrograde-injected through the suprahepatic vein at 20 mL/s. The liver was maintained watertight for 5 min. After this time, the clamps were removed and the organ was ready to be implanted.

#### 4.2.3. Recipient

Classic hepatectomy without inferior vena cava preservation was performed, as previously described. In brief, the liver hilum elements were dissected following this order: bile duct, right and left hepatic arteries, and portal vein. Then, the triangular and gastrohepatic ligaments were sectioned to completely liberate and mobilize the liver. Both the supra- and infrahepatic inferior vena cava were dissected. Then, the cava, portal, and hepatic veins were clamped. The elements of the pedicle, hepatic veins, and cava were sectioned and the liver was removed. This anhepatic stage had to be rapid and it did not last more than 20–25 min since this would compromise the safety of the procedure and the recipients’ survival. The graft implant phase started with the anastomosis of the donor’s and recipient’s suprahepatic inferior vena cava. Then, the portal vein was anastomosed and these vessels were declampled to reduce intestinal ischemia and favor venous return. Subsequently, anastomosis of the infrahepatic vena cava was performed and the clamps were removed, thus improving the hemodynamic status of the pig due to complete venous reperfusion. Then, the hepatic artery was anastomosed, controlling its length in order to be neither twisted nor redundant. Finally, the bile duct was also anastomosed with the aid of a silicone tube inserted as an internal tutor. Then, the abdominal cavity was cleaned and the abdominal wall was closed.

### 4.3. Post-Transplantation Care

Pigs were monitored every 8 h from intervention until sacrifice, evaluating their stance, breathing, food intake, and response to stimuli. The medication described above was administered during the stabling period through the 3-way catheter placed in the jugular vein.

### 4.4. Plasmid

Plasmid p2F-*hIL-10* (6.86 Kb), containing human IL-10 cDNA driven by the pCMV promoter, was constructed by cloning IL-10 into the HindIII site of pVITRO2 (Invitrogen, Thermo Fisher, Waltham, MA, USA).

### 4.5. Biochemical Parameters and Liver Enzymes Determination

Blood biochemical parameters were quantified ‘in situ’ using the i-STAT 1^®^ blood analyzer (Abbott Laboratories, Chicago, IL, USA) during the intervention and the entire period of stabling from the jugular vein. Two different analysis cartridges were employed: CG4+ (lactate, pH, pCO_2_, pO_2_, TCO_2_, HCO_3_, sO_2_, base excess or BE); and CG8+ (sodium, potassium, ionized calcium, glucose, hematocrit, hemoglobin). Plasma GOT and GPT were quantified post-intervention by an external laboratory employing an electrochemical method.

### 4.6. Quantitative PCR and RT-PCR

Tissue samples representing the whole liver were collected after sacrifice and homogenized in buffer (Promega^®^, Madison, WI, USA) using an Ultra-Turrax homogenizer (Hielscher Ultrasonics GmbH, Teltow, Germany). Further purifications were performed using the Maxwell RNA Purification From Tissue Kit (Promega^®^, Madison, WI, USA) before spectrophotometric quantification. RNA retrotranscription to cDNA was carried out using the High Capacity cDNA Archive Kit (Thermo Fisher^®^, Waltham, MA, USA). For real-time qPCR, TaqMan PCR Master Mix (Thermo Fisher^®^, Waltham, MA, USA) was employed according to the instructions of the manufacturer. The specific oligonucleotides for the human IL-10 employed were from a pre-mixed kit from Thermo Fisher (cat no. Hs00961622_m; Thermo Fisher^®^, Waltham, MA, USA).

### 4.7. Cytokine Determination

#### 4.7.1. hIL-10 ELISA

Blood samples were collected at 1 h, 6 h, 1 day, 2 days, 4 days, 7 days, and 10 days after gene transfer. Plasma was separated and the protein presence was determined. For human IL-10 quantification, the BD OptEIA^®^ Human IL-10 ELISA Set (Becton and Dickinson Biosciences, Franklin Lakes, NJ, USA) was used, following the manufacturer’s instructions.

#### 4.7.2. Determination of Porcine Cytokines

Blood samples were collected at 1 h, 2 days, 4 days, and 10 days post-transplantation. Cytokine concentrations in plasma were quantified using a Magpix Multiplex analyzer (Luminex, Austin, TX, USA), based on fluorescence microscopy and the flow cytometry technique. We simultaneously determined IL-1β, IL-4, IL-6, IL-8, IL-10, IL-12–23, 40p, IFN-α, IFN-γ, and TNF-α, employing the Cytokine & Chemokine 9-Plex Porcine ProcartaPlex™ (Invitrogen, Waltham, MA, USA), following the manufacturer’s instructions. TGF-β was separately quantified using the TGF beta 1 Porcine ProcartaPlex™ Simplex Kit (Invitrogen, Waltham, MA, USA).

### 4.8. Anatomopathological Evaluation of Tissue Samples

After the stabling period, animals were sacrificed and tissue samples (2 cm × 1 cm × 2 mm) from all liver lobes were collected and preserved in formaldehyde. Tissue slides were stained with hematoxylin-eosin and observed under an optical microscope at 10× and 20× magnification. Different injury signs were evaluated and classified as intervention-derived parameters, such as ischemic lesions and general inflammation traces, and organ rejection, defined by the Banff triad (duct injury, portal inflammation, and venular inflammation). A Liver Transplantation Hospital anatomopathologist assigned a severity value from 0 to 3 to each injury sign in each sample in a blind study. The values were aggregated and averaged for each animal to establish its overall damage, separating intervention-derived injury from rejection.

### 4.9. Statistical Analyses

Results obtained in the transfected and control groups were compared statistically with a paired *t*-test, employing Prism 5 software (Graphpad Software, Carlsbad, CA, USA), and *p*-value was considered significant from 0.05 downwards. Data graphing was performed using the same software.

## 5. Conclusions

In conclusion, our study supports that hydrodynamic transfer of the naked human IL-10 gene in the liver transplantation setting is a safe, viable, and efficient procedure for achieving sustained plasma levels of human protein with potential functional activity. This increased protein expression was associated with elevated levels of both pro- and anti-inflammatory cytokines, which did not lead to significant liver tissue injury. Moreover, we observed elevated plasma levels of some cytokines known for their capacity to mediate immunotolerance activity, such as TGF-β and IL-6, which could facilitate the graft tolerance process. In this regard, the pharmacokinetics (production and release to bloodstream of interleukin-10) of the procedure were verified and when translated to humans, the pharmacodynamics (drug-receptor) process would require additional dose–response (hydrodynamic gene transfer efficacy is dose-dependent and the doses employed in this study could be largely augmented) and longer-term studies.

## Figures and Tables

**Figure 1 ijms-25-01491-f001:**
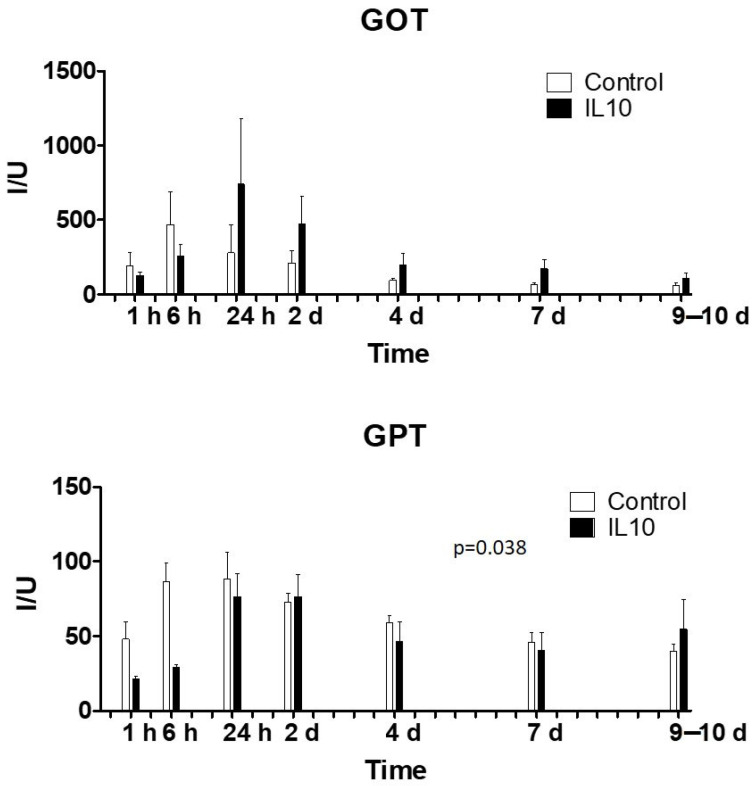
Liver enzymes in plasma after liver transplant. GOT: Glutamate-oxaloacetate transaminase or aspartate aminotransferase; GPT: Glutamic-pyruvic transaminase or alanine aminotransferase. *p*-value GOT = 0.35; *p*-value GPT = 0.24. *N*: 5 pigs per group. Dose: 20 µg/mL; 200 mL; 20 mL/s.

**Figure 2 ijms-25-01491-f002:**
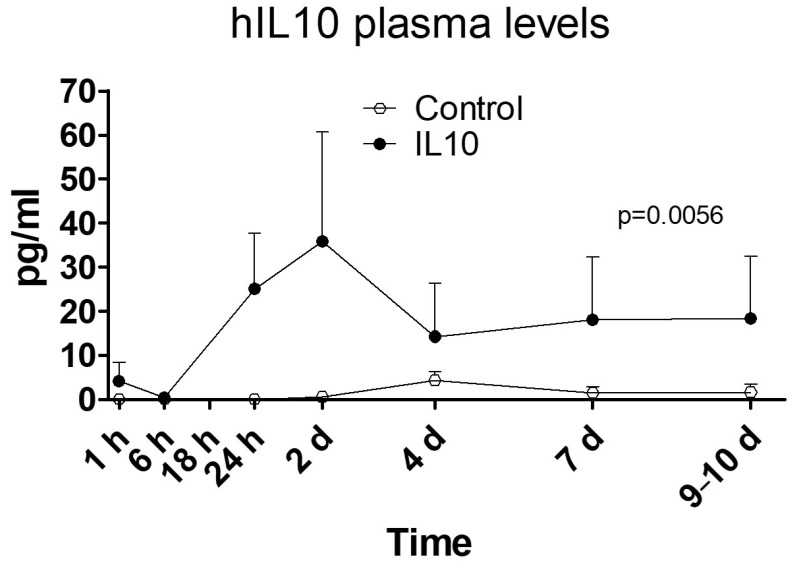
hIL-10 protein in plasma. Protein was quantified by ELISA in plasma from blood samples obtained during the follow-up. *N*: 5 pigs per group. Dose: 20 µg/mL; 200 mL; 20 mL/s.

**Figure 3 ijms-25-01491-f003:**
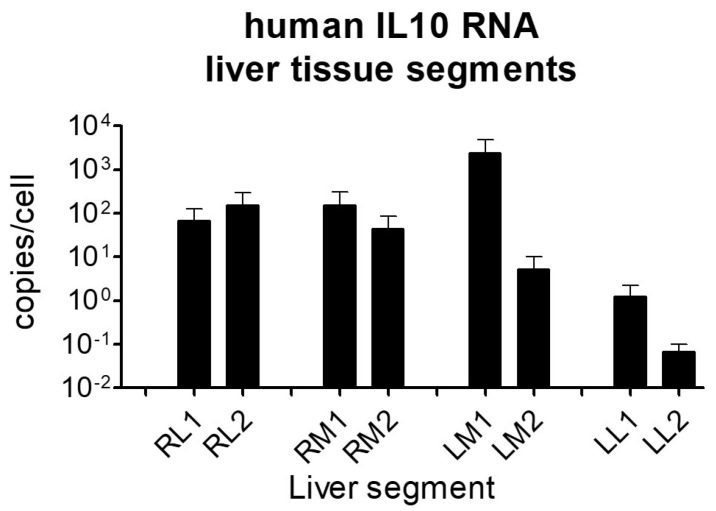
*hIL-10* RNA tissue expression. Tissue samples from 8 different liver areas were collected and the RNA was extracted and retrotranscribed. *hIL-10* cDNA was and quantified by qPCR and expressed as copies per cell. The unspecific values obtained in control animal samples were considered as the threshold and subtracted from the levels obtained in samples from transfected animals. RL: right lateral; RM: right medial; LM: left medial; LL: left lateral; 1: proximal; 2: distal. *N*: 5 pigs per group. Dose: 20 µg/mL; 200 mL; 20 mL/s.

**Figure 4 ijms-25-01491-f004:**
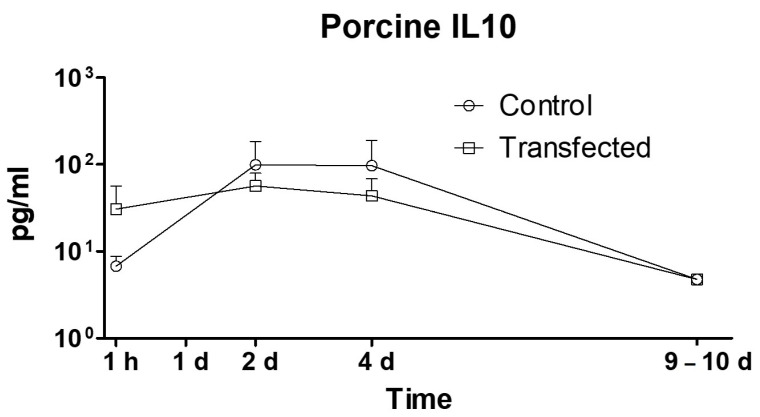
Porcine IL-10 protein in plasma. Protein was quantified in plasma from blood samples obtained during the follow-up using a Magpix Multiplex analyzer, based on fluorescence microscopy and flow cytometry. Levels achieved in control and treated animals were not significantly different (*p* = 0.39; *t*-test). *N*: 5 pigs per group. Dose: 20 µg/mL; 200 mL; 20 mL/s.

**Figure 5 ijms-25-01491-f005:**
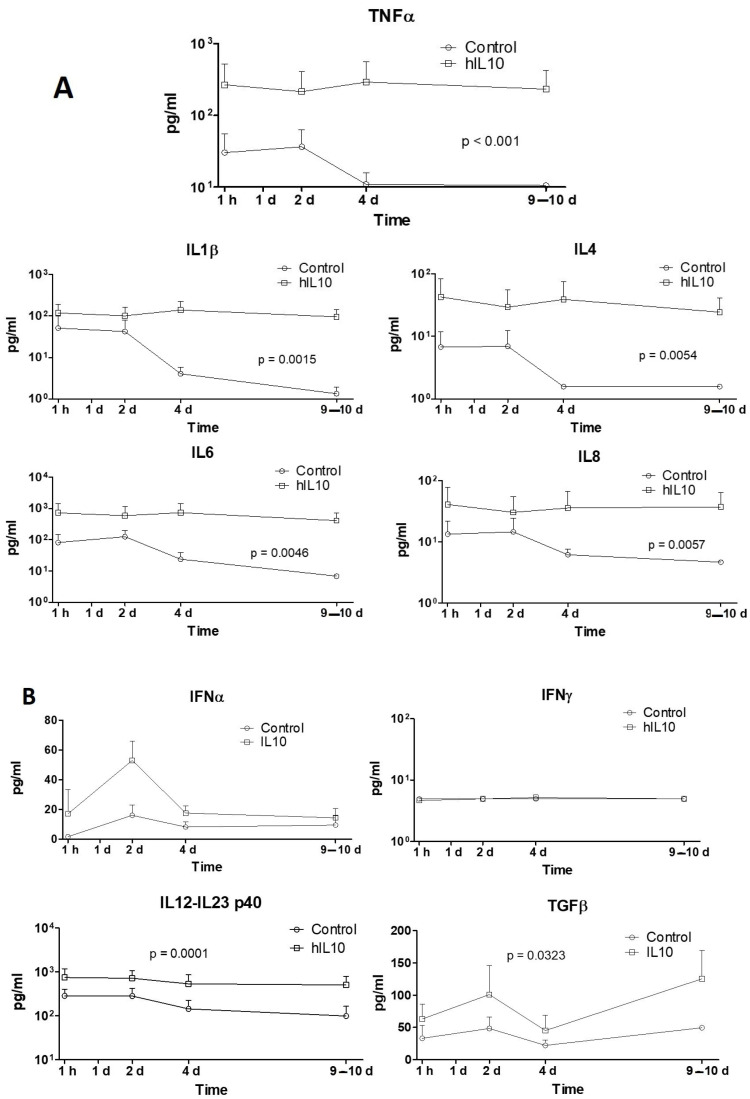
(**A**) Plasma levels of porcine cytokines TNFα, IL1β, IL4, IL6 and IL8. (**B**) The levels of IFNα, IFNγ, IL12-IL23 p40 and TGFβ. Proteins were quantified in plasma from blood samples obtained at different time points during the follow-up using a Magpix Multiplex analyzer, based on fluorescence microscopy and flow cytometry. Statistical test: *t*-test. *N*: 5 pigs per group. Dose: 20 µg/mL; 200 mL; 20 mL/s.

**Figure 6 ijms-25-01491-f006:**
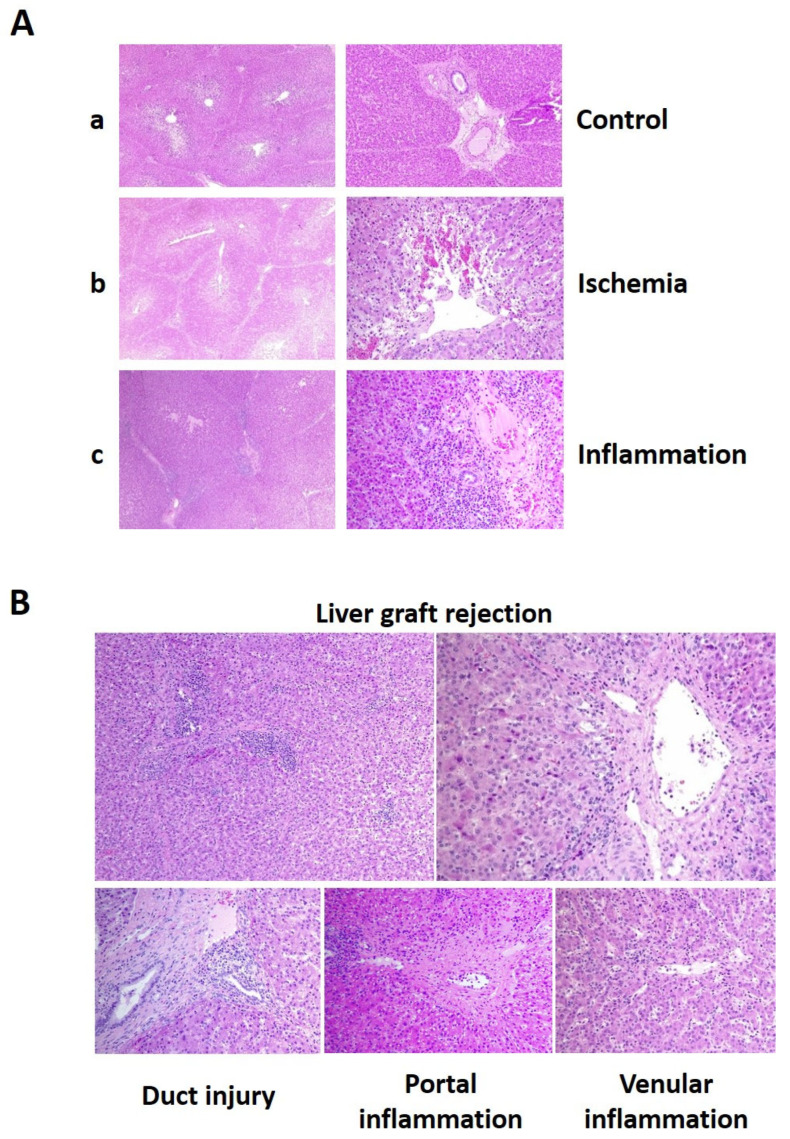
Images of general injury (**A**) and signs of acute rejection (**B**) in liver tissue samples under a bright microscope at 10× and 20× magnification after hematoxylin-eosin staining. In panel A, left pictures were taken at 10× magnification and right pictures at 20×. (**A**) a images represent healthy control tissue without damage; b pictures show a tissue sample with ischemia and c image shows an area of liver sample with inflammation and cell infiltration. Superior left picture in panel (**B**) shows a general view of tissue sample (10×) from a liver with rejection signs. Other images are details (20×) of the sample showing typical Banff rejection signs.

**Figure 7 ijms-25-01491-f007:**
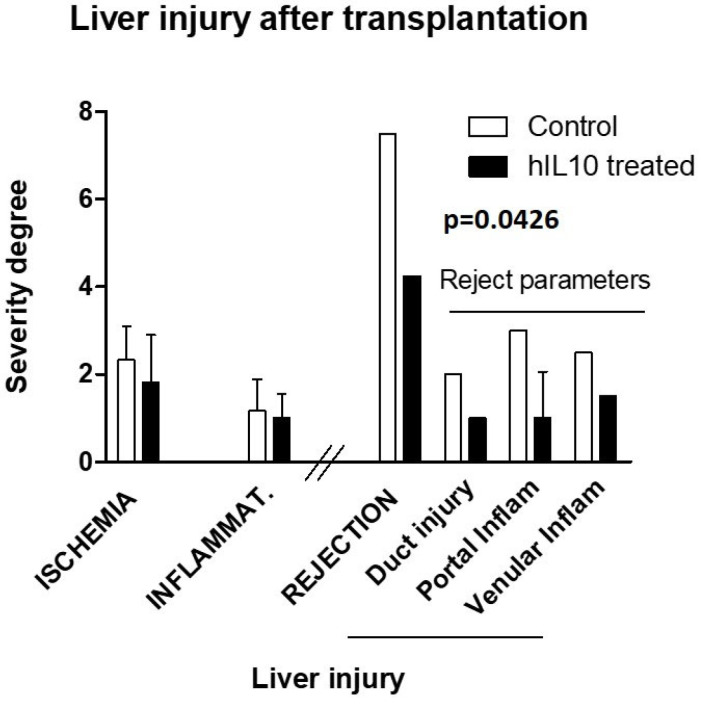
Graph representing the general injury and acute rejection degree of severity in liver tissue samples assigned by a blinded pathologist. Each of these signs received a value from 0 to 3 depending on their severity in each individual sample. These were aggregated and averaged for each animal to establish its overall damage. Statistical test: *t*-test. *N*: 1 pig per group presenting rejection signs). Dose: 20 µg/mL; 200 mL; 20 mL/s.

**Figure 8 ijms-25-01491-f008:**
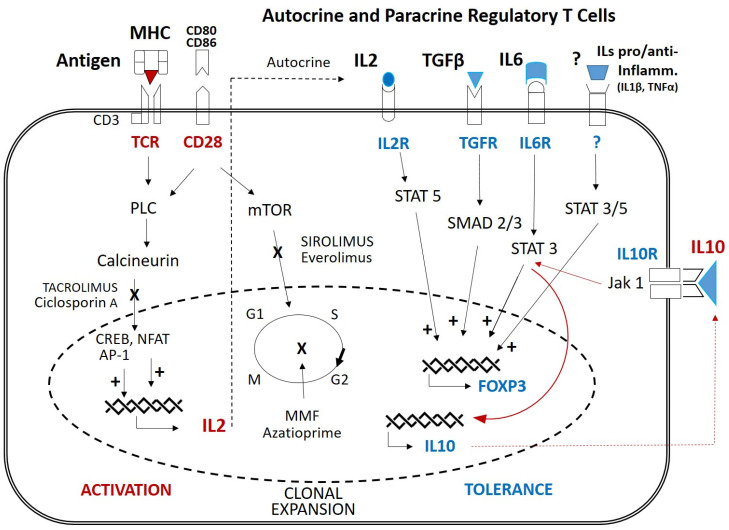
T cell activation/tolerance and mechanisms involved. MHC: major histocompatibility complex; TCR: T cell receptor; CD: cluster of differentiation; MMF: mycophenolate mofetil; PLC: phospholipase C; NFAT: nuclear factor of activated T cells; CREB: CAMP response element-binding protein; AP-1: activator protein 1; STAT: signal transducer and activator of transcription; SMAD: suppressor of mothers against decapentaplegic; mTOR: mammalian target of rapamycin; Jak: Janus kinase.

**Table 1 ijms-25-01491-t001:** Biochemical parameters in plasma after liver transplant. *N*: 5 pigs per group. Dose: 20 µg/mL; 200 mL; 20 mL/s.

	pH					Creatinine (mg/dL)				Lactate (mmol/L)				Hemoglobin (g/dL)			
	Control			IL-10-Treated		Control		IL-10-Treated		Control		IL-10-Treated		Control		IL-10-Treated	
	Average		SD	Average	SD	Average	SD	Average	SD	Average	SD	Average	SD	Average	SD	Average	SD
Pre-surgery	7.39		0.04	7.06	0.50	1.20	0.14	ND	ND	1.18	0.32	0.57	0.00	8.85	1.91	ND	0.00
1 d	7.49		0.05	7.44	0.07	1.70	0.10	1.33	0.15	1.15	0.53	1.44	0.21	8.60	1.39	9.14	0.93
2 d	7.52		0.03	7.48	0.01	1.25	0.07	1.15	0.17	0.79	0.17	0.72	0.18	7.80	0.00	8.30	1.04
4 d	7.47		0.05	7.48	0.01	1.13	0.10	1.17	0.06	0.77	0.19	1.13	0.32	8.30	0.81	9.88	1.96
7 d	7.49		0.03	7.48	0.04	1.15	0.31	1.05	0.06	1.01	0.48	1.17	0.34	7.93	1.12	10.40	2.08
9–10 d	7.42		0.10	7.41	0.08	1.07	0.21	0.97	0.15	1.84	1.44	1.27	0.73	8.27	0.50	9.63	1.40
	**Bilirubin (mg/dL)**				**Glucose (mg/dL)**				**Albumin (g/L)**								
	**Control**		**IL-10-Treated**		**Control**		**IL-10-Treated**		**Control**			**IL-10-Treated**					
	Average	SD	Average	SD	Average	SD	Average	SD	Average		SD	Average	SD				
1 d	0.46	0.18	0.33	0.13	88.25	9.50	104.60	31.39	25.50		5.80	25.80	2.86				
2 d	0.35	0.24	0.47	0.18	101.00	37.42	88.40	11.63	24.20		1.92	28.20	2.28				
4 d	0.50	0.33	0.64	0.36	87.00	10.93	63.60	34.34	24.60		2.51	26.20	2.05				
7 d	0.54	0.28	0.84	0.64	83.60	9.34	52.00	37.40	23.80		3.56	25.20	2.77				
9–10 d	0.78	0.55	0.91	0.65	78.67	17.16	69.50	20.86	25.67		2.52	28.75	5.25				

## Data Availability

Data are contained within the article.

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
