# Peer review of "Safe Procedure for Efficient Hydrodynamic Gene Transfer to Isolated Porcine Liver in Transplantation"

_ijms, 2024, doi:10.3390/ijms25031491_

Round 1

Reviewer 1 Report

Comments and Suggestions for Authors

The manuscript entitled “Safe procedure for efficient hydrodynamic gene transfer to isolated porcine liver in transplantation” by Luis Sendra et al. studied potential effect of hydrodynamic injection of hIL10 gene on safety, expression efficacy, and the immediate post-transplantation expression of cytokines in an orthotopic liver transplantation porcine. The work presented herein shown certain importance. However, there are some issues need to be addressed before publishing in International Journal of Molecular Sciences.

1.     In this manuscript, the authors used hydrodynamic injection of naked DNA. However, it will be easier for the readers to understand if the authors could provide a brief introduction of the advantage of Hydrodynamic gene transfer compared to other gene transfection methods in the manuscript?

2.     In this introduction, the authors are encouraged to detailed introduce the relationship between IL10 and immunosuppressant in organ transplantations.

3.     In this introduction, the authors are encouraged to detailed the differences between human IL10 and porcine IL10.

4.     In this introduction, it will be better if authors could cite the latest literature.  

5.     In page 3 line 113, it is better to give the full name of GPT at the beginning of the paper, as well as GOT, IFNα, IL1β, TNFα, IL6, IL8, IL4 and TGFβ.  

6.     In all table and figures, it is better to give the number of pigs and dose?  

7.     In the Figure 3, human IL10 RNA in liver tissue segments should be evaluated.  

8.     In the Figure 5, it will be better if authors give a more detailed explanation of the increased expression of IFNα and IL12 after transfected hIL10.  

9.     In page 9, it will be easier for the readers to understand if the authors could provide a brief introduction of mechanism of IL10 regulating T cell activation/tolerance. 

10.  In reference part, format of ref11 and ref29 needs to be consistent with the other references.

Author Response

The manuscript entitled “Safe procedure for efficient hydrodynamic gene transfer to isolated porcine liver in transplantation” by Luis Sendra et al. studied potential effect of hydrodynamic injection of hIL10 gene on safety, expression efficacy, and the immediate post-transplantation expression of cytokines in an orthotopic liver transplantation porcine. The work presented herein shown certain importance. However, there are some issues need to be addressed before publishing in International Journal of Molecular Sciences.

Firstly, we would like to thank the referee for the effort in reviewing the manuscript and for the suggestions that we try to address within the new version of the manuscript and we are sure will improve the overall quality of our work.

  1. In this manuscript, the authors used hydrodynamic injection of naked DNA. However, it will be easier for the readers to understand if the authors could provide a brief introduction of the advantage of Hydrodynamic gene transfer compared to other gene transfection methods in the manuscript?

We appreciate the reviewer’s suggestion. We have added this information within the manuscript by including the following paragraph: ‘Hydrodynamic procedure for gene transfer consists in the rapid administration of a large volume of saline aqueous solution containing the gene of interest at defined concentrations. This strategy presents the advantage that DNA is administered without any carrier and then neither toxicities, more common with viral vectors [1], nor interactions can occur and the gene release in the target is not limited [2]. As disadvantages, the gene is delivered without targeting so this could potentially access any cell and the volume requirement for efficient gene transfer could compromise the patient´s hemodynamics. The strategy developed in previous works [3] and the present one for loco-regional administration of gene solution in vascular isolated organs per-mitted circumvent, partially at least, these disadvantages since the DNA was injected only in the target organ and the conditions for administration could be moderated. Furthermore, the technique would permit the administration of gene using minimally invasive catheterization, making its clinical translation feasible [4].’

  1. In this introduction, the authors are encouraged to detailed introduce the relationship between IL10 and immunosuppressant in organ transplantations.

According to the reviewer comment, we have detailed it in the new version by adding the following paragraph in introduction section: ‘For this reason, it was decided to employ a gene that encoded a protein with immunomodulatory function, as interleukin 10, instead of a tracer gene, as GFP, since this cytokine plays a central role in the modulation of immune activation through different pleiotropic functions. Interleukin-10 (IL-10) is a pleiotropic cytokine that has a fundamental role in modulating inflammation and in maintaining cell homeostasis. It primarily acts as an anti-inflammatory cytokine, protecting the body from an uncontrolled immune response. Given the pivotal role of IL-10 in immune modulation, this cytokine could have relevant implications in pathologies characterized by hyperinflammatory state. IL-10 leads to the shutdown of the inflammatory immune response, both directly, by the suppression of macrophages and dendritic cells activity, and indirectly, by limiting T cells activation, differentiation and effector function and promot-ing peripheral tolerance [23].’

  1. In this introduction, the authors are encouraged to detailed the differences between human IL10 and porcine IL10.

Thanks for the suggestion, we have added the following paragraph to detail these differences: ‘This molecule was reported to improve the outcome in transplantation [24] and polymorphisms in its gene have been described to influence the transplantation success. Additionally, we selected it because it could also serve as tracer molecule at RNA and protein levels that permit differentially quantify the expression of the exogenous human gene. It has been reported that the amino acid sequence homology between human and mouse, rabbit, pig and cow IL10 is high. Human and porcine IL10 proteins present a 76% homology, more than the observed between human and mouse proteins. Human IL10 cDNA presents important homology with pig IL10, 82% in nucleotides sequence [25]. Moreover, IL10 has been shown to be functionally active across species barriers, including human, mouse, rabbit, pig and cow [26-28].’

  1. In this introductionit will be better if authors could cite the latest literature.  

We have included more recent references

  1. In page 3 line 113, it is better to give the full name of GPT at the beginning of the paper, as well as GOT, IFNα, IL1β, TNFα, IL6, IL8, IL4 and TGFβ. 

The reviewer is right, we should have introduced it. We provide now the complete name of these molecules within the new version of the manuscript.

  1. In all table and figures, it is better to give the number of pigs and dose? 

We have modified them according to the reviewer’s comment

  1. In the Figure 3, human IL10 RNA in liver tissue segments should be evaluated.

We are not sure if we fully understand the referee’s suggestion since the amount of human IL10 mRNA in each liver sample was quantified and it is represented in figure 3 as copies per cell but we have now commented the results more thoroughly. We have added the following paragraph: ‘Figure 3 shows the tissue amount of human IL10 RNA in the different areas of the liver, expressed as copy number per cell after subtracting the background read. All liver lobes presented detectable expression of hIL10 mRNA, although the yield was different depending on the area of the organ. Whereas both medial and lateral segments of right lobes showed similar intermediate transcription rate, in left lobes, great differences between medial and lateral areas were detected, the latter showing the lowest efficacy. The proximal area of medial left lobe showed the highest production of mRNA, more than 1,000-fold higher than in lateral left segments. We think that the reason for such differences must be the anatomical structure of suprahepatic vasculature that orientates the catheter in that direction and favors the irrigation of those areas’. 

  1. In the Figure 5, it will be better if authors give a more detailed explanation of the increased expression of IFNα and IL12 after transfected hIL10. 

Thanks for the suggestion. We had partially included this information within the discussion but we consider that the reviewer is right and it could better clarify the result. For this reason, it has been addressed within the results section in the new version of the manuscript, by adding the following sentence: ‘Although it was not significant (p=0.089), the expression of IFNα increased in trans-fected group during the first 2 days maybe due to the administration of double-stranded DNA, as previously reported. Nucleic acids derived from pathogens or released from damaged cells can activate the immune system.TLR9 membrane-bound DNA sensor detects CpG DNA at the endosome to induce type I IFN production. There are also TLR9-independent pathways that can recognize double-stranded DNA (dsDNA) in the cytoplasm, preferentially mediating the production of IFN-α but not IL-12 [29, 30]. Interleukin 12 (IL12) expression levels in transfected pigs was significantly higher (p=0.0001) during the follow-up and this could be due to an immune response against ssRNA presence in bloodstream what could happen since the plasmid administered has already proved to express the hIL10 mRNA in few hours [17, 19]. RNA-sensing mechanisms are mediated by Toll-like receptors (TLR) and by retinoic acid-inducible gene I-like receptors (RLRs) and induce the secretion of IL-12 but not IFN-α [29, 30].’

  1. In page 9, it will be easier for the readers to understand if the authors could provide a brief introduction of mechanism of IL10 regulating T cell activation/tolerance.

We appreciate this suggestion. We have added the following paragraph accordingly: ‘Interleukin 10 (IL10) is a pleomorphic cytokine produced by most activated immune cells, including B cells, mast cells, granulocytes, macrophages, dendritic cells, and mul-tiple T cell subsets, as Tregs. Receptor ligation activates JAK/STAT3 signaling, leading to changes in the expression of immunomodulatory genes, such as Foxp3, inhibit pro-inflammatory mediators, decrease antigen presentation and enhance the tolerance functions of these cells. IL10 can act as feedback regulator that affects the control and resolution of inflammation via autocrine and paracrine mechanisms. In addition, IL10 is thought to inhibit apoptotic signaling pathways, such as the p38 MAPK (mito-gen-activated protein kinase) pathway, and maintain and expand Treg population [37]. On the other side, IL-10 can positively enhance activation and proliferation of certain immune cell types, including mast cells, CD8+ T cells, NK cells, and B cells, although the molecular mechanisms and functional consequences of such activity remain to be elucidated [38].’

  1. In reference part, format of ref11 and ref29 needs to be consistent with the other references.

Despite reviewing it, we could not find the inconsistency in reference format. We checked the references at https://pubmed.ncbi.nlm.nih.gov/ and included them literally within the new version of the manuscript

Reviewer 2 Report

Comments and Suggestions for Authors

Overall the authors present an interesting paper with 2 major findings. First of all, they show a safe and viable transfer of naked human IL-10 gene in a setting of liver transplantation. This part of the manuscript is flawless with adequate data.

The second part of this study shows cytokine responses in this procedure and an overall response to liver injury/rejection. This section also provides many data, which I believe should be better adressed in results and discussion section. To be more precise:

1. In table 1 and figure 1 p values should be provided. T.ex. the authors discuss the higher GPT levels in IL-10 treated liver after 6 hours but not GOT levels after 24 hours. If the reason for this is the low p value for GOT levels between 2 groups this should be provided.

2. Also in results section, left medial lobe seems to yield the highest expression. A possible explanation should be given for that

3. In figure 7, the p value is for the overall score or for each one of duct injury, portal inflammation and venular inflammation? Moreover, this finding should be discussed more in discussion section, even though it presents a very intrsting finding. A possible scenario for that should be given (for help see also Assadiasl S, Cytokines 2021)

4. Lastly, in figure 8, the action of IL-10 in T-cells should also be presented.  

Author Response

Overall the authors present an interesting paper with 2 major findings. First of all, they show a safe and viable transfer of naked human IL-10 gene in a setting of liver transplantation. This part of the manuscript is flawless with adequate data.

The second part of this study shows cytokine responses in this procedure and an overall response to liver injury/rejection. This section also provides many data, which I believe should be better adressed in results and discussion section. To be more precise:

We would like to thank the referee for the comments, which we try to address in the new version of the manuscript and we are sure will improve the overall quality of this work.

  1. In table 1 and figure 1 p values should be provided. T.ex. the authors discuss the higher GPT levels in IL-10 treated liver after 6 hours but not GOT levels after 24 hours. If the reason for this is the low p value for GOT levels between 2 groups this should be provided.

Thanks to this comment we realized that we had made a mistake reporting the GPT results, the p-value did not correspond with the statistical analysis. None of the parameters proved significant difference. We have now corrected it (p-value 0.24) and added the p-value corresponding to GOT (p-value 0.35) data analysis within the table caption. We are really sorry for this mistake

  1. Also in results section, left medial lobe seems to yield the highest expression. A possible explanation should be given for that

Thanks for the suggestion, we have now commented the results more thoroughly. We have added the following paragraph: ‘Figure 3 shows the tissue amount of human IL10 RNA in the different areas of the liver, expressed as copy number per cell after subtracting the background read. All liver lobes presented detectable expression of hIL10 mRNA, although the yield was different depending on the area of the organ. Whereas both medial and lateral segments of right lobes showed similar intermediate transcription rate, in left lobes, great differences between medial and lateral areas were detected, the latter showing the lowest efficacy. The proximal area of medial left lobe showed the highest production of mRNA, more than 1,000-fold higher than in lateral left segments. We think that the reason for such differences must be the anatomical structure of suprahepatic vasculature that orientates the catheter in that direction and favors the irrigation of those areas’

  1. In figure 7, the p value is for the overall score or for each one of duct injury, portal inflammation and venular inflammation? Moreover, this finding should be discussed more in discussion section, even though it presents a very interesting finding. A possible scenario for that should be given (for help see also Assadiasl S, Cytokines 2021)

The p-value correspond to the comparison of the overall score of rejection parameters. Following the relevant suggestion of the reviewer, we have added the following paragraph to discuss these findings: ‘Despite the increased and sustained expression of both pro- and an-ti-inflammatory cytokines after hIL10 transfection, the hIL10 treated animals generally displayed better well-being, as observed through their behavior and response to stim-uli. In accordance with the levels of analyzed cytokines, one would anticipate that pigs treated with IL10 would exhibit a greater and sustained inflammatory response throughout the entire study duration. Mediators conventionally reported as pro-inflammatory and, consequently, of potential risk for transplant evolution [32] (IL1, IL6, IL8, TNFα, IL12-IL23) maintained higher levels throughout the analysis pe-riod in the treated pigs compared to the controls, where they normalized by days 2-4 post-intervention. However, assessment of anatomopathological parameters indicated that the severity of Banff parameters, used to define the onset of rejection, was signifi-cantly higher (p=0.0426) in the case of the control group. Conversely, levels of proteins with potential anti-inflammatory roles (IL10, TGFb) were also elevated in treated pigs, which could favor the establishment of tolerance. These findings would require con-firmation in longer-term studies where Treg response could be assessed. These plei-otropic cytokines are known to participate in various activation or modulation [33, 34] pathways during the acute phase of inflammation.’

  1. Lastly, in figure 8, the action of IL-10 in T-cells should also be presented. 

According to the reviewer suggestion, we have included the available information upon the direct action of IL10 on Treg through Jak1/STAT3 signaling pathway. Additionally, we have included the following sentence to explain the new information: ‘The direct action of IL10 on Treg through Jak1/STAT3 signaling pathway could be the responsible for enhancing the autocrine production of IL10 and this control the feed-back response and possibly the expansion of Treg cells [43].’

Round 2

Reviewer 1 Report

Comments and Suggestions for Authors

The revised manuscript has been addressed the concerns and questions from the reviewers. So it is acceptable for publishing in International Journal of Molecular Sciences.

Reviewer 2 Report

Comments and Suggestions for Authors

I think that after the corrections in the manuscript by the authors, the manuscript is now ready for publication

Comments on the Quality of English Language

Minor editing required